# Oxidative Stress Response of Probiotic Strain *Bifidobacterium longum* subsp. *longum* GT15

**DOI:** 10.3390/foods12183356

**Published:** 2023-09-07

**Authors:** Olga V. Averina, Aleksey S. Kovtun, Dilara A. Mavletova, Rustam H. Ziganshin, Valery N. Danilenko, Dasha Mihaylova, Denica Blazheva, Aleksandar Slavchev, Mariya Brazkova, Salam A. Ibrahim, Albert Krastanov

**Affiliations:** 1Vavilov Institute of General Genetics, Russian Academy of Sciences, 119991 Moscow, Russia; kovtunas25@gmail.com (A.S.K.); mavletova@vigg.ru (D.A.M.); valerid@vigg.ru (V.N.D.); 2Shemyakin-Ovchinnikov Institute of Bioorganic Chemistry, Russian Academy of Sciences, 117997 Moscow, Russia; ziganshin@mail.ru; 3Department of Biotechnology, University of Food Technologies, 4002 Plovdiv, Bulgaria; dashamihaylova@yahoo.com (D.M.); a_krastanov@uft-plovdiv.bg (A.K.); 4Department of Microbiology, University of Food Technologies, 4002 Plovdiv, Bulgaria; d_blazheva@uft-plovdiv.bg (D.B.); a_slavchev@uft-plovdiv.bg (A.S.); 5Food Microbiology and Biotechnology Laboratory, Food and Nutritional Science Program, North Carolina A&T State University, Greensboro, NC 27411-1064, USA; ibrah001@ncat.edu

**Keywords:** oxidative stress, bifidobacteria, genomic, transcriptomic, proteomic and metabolomic analyses

## Abstract

*Bifidobacterium* is a predominant and important genus in the bacterial population of the human gut microbiota. Despite the increasing number of studies on the beneficial functionality of bifidobacteria for human health, knowledge about their antioxidant potential is still insufficient. Several in vivo and in vitro studies of *Bifidobacterium* strains and their cellular components have shown good antioxidant capacity that provided a certain protection of their own and the host’s cells. Our work presents the data of transcriptomic, proteomic, and metabolomic analyses of the growing and stationary culture of the probiotic strain *B. longum* subsp. *longum* GT15 after exposure to hydrogen peroxide for 2 h and oxygen for 2 and 4 h. The results of the analysis of the sequenced genome of *B. longum* GT15 showed the presence of 16 gene-encoding proteins with known antioxidant functions. The results of the full transcriptomic analysis demonstrated a more than two-fold increase of levels of transcripts for eleven genes, encoding proteins with antioxidant functions. Proteomic data analysis showed an increased level of more than two times for glutaredoxin and thioredoxin after the exposure to oxygen, which indicates that the thioredoxin-dependent antioxidant system may be the major redox homeostasis system in *B. longum* bacteria. We also found that the levels of proteins presumably involved in global stress, amino acid metabolism, nucleotide and carbohydrate metabolism, and transport had significantly increased in response to oxidative stress. The metabolic fingerprint analysis also showed good discrimination between cells responding to oxidative stress and the untreated controls. Our results provide a greater understanding of the mechanism of oxidative stress response in *B. longum* and the factors that contribute to its survival in functional food products.

## 1. Introduction

Functional foods are traditional or novel foods or dietary components that are considered to have health benefits in addition to providing basic nutrition. Thus, functional foods can help to reduce the prevalence of diseases by optimizing and regulating the capacity of the human microbiome and immune system to prevent and control both infections caused by pathogens, and pathologies resulting from functional alterations in the host. *Bifidobacterium* is one of the important and predominant bacterial genera of the human gut microbiota. Their positive functional roles for human health are well documented [1]. The amount of bifidobacteria in the gut microbiota of vaginally delivered breast-fed infants reaches 90%. During the later stages of life, it is reduced to 5%, and even less in elderly people [2].

Bifidobacteria are high-G+C Gram-positive microorganisms belonging to the subclass *Actinobacteridae* of phylum Actinobacteria. Like other colon bacteria, bifidobacteria are strict anaerobes, so the presence of oxygen is an important stress factor for them. Exposure to oxygen can lead to the accumulation of reactive oxygen species (ROS), which is lethal to cells. ROS leads to protein misfolding and aggregation, DNA damage, and lipid peroxidation. Several in vivo and in vitro studies have demonstrated great antioxidant capacity of certain strains of the genus *Bifidobacterium* and their components, which provides a certain degree of protection from oxidative damage for both the bacterial and host cells [3]. Bifidobacteria can exert antioxidative (AO) activity via various mechanisms: synthesis of AO enzymes, peptides, and thiols; compounds with AO properties, and chelation of toxic ions (Fe^2+^ and Cu^2+^). AO functions are strain-specific [4,5,6]. Strains of the dominant species of bifidobacteria, *Bifidobacterium longum* subsp. *Longum,* are oxygen-sensitive (grow in the presence of 5% O_2_ in liquid culture) [7]. Additionally, strains of *B. longum* could grow in L-shaped tubes with occasional shaking [8]. Genome sequence analysis has indicated that the genomes of *B. longum* strains do not have genes encoding NADH peroxidase or superoxide dismutase (SOD), although they do have genes for NADH oxidase [9], alkyl hydroperoxide reductase (AhpC) [10], thioredoxin reductase [11], and other AO enzymes [3]. 

Bifidobacteria, due to their health-promoting properties in humans, are considered probiotics. In recent years, there has been an increasing interest in bifidobacteria as an accompanying drug for the treatment and prevention of various diseases, including neurological diseases, accompanied by the development of oxidative stress (OS). Successful commercial application of probiotic bifidobacteria depends on their ability to survive during industrial production and storage [12]. ROS damage is one of the main reasons for the loss of viability of anaerobic probiotics such as bifidobacteria. Current knowledge about the mechanism of antioxidant protection of bifidobacteria is still insufficient. In this regard, more attention should be paid to the study of the antioxidant potential of bifidobacteria, especially in the more common and dominant species inhabiting the human intestine, such as *B. longum* subsp. *longum*. Different omics technologies for the analysis of probiotic strains of bifidobacteria allow detecting the intrinsic defense systems that protect cells from OS.

With modern biotechnology, the levels of important biologically active and functional compounds in food can be improved. For instance, advancements in the fields of genomics, proteomics nutrigenomics, and metabolomics could be applied in order to enhance the production of functional compounds in food products as functional foods. Our work presents the data from genomic, transcriptomic, proteomic, and metabolomic analyses of a probiotic strain of *B. longum* subsp. *longum* GT15 after exposure to hydrogen peroxide for 2 h and oxygen for 2 and 4 h. This strain was isolated from the stool of a healthy volunteer from the Moscow region (Russia). The full-length genome sequence of this strain has been completed (GenBank accession no. CP006741) [13]. The immunomodulatory properties of *B. longum* GT15 have previously been shown [14] and its AO properties are described in this work. This allows recommending its application in functional products aimed at reducing the inflammatory process accompanying various diseases.

To understand the mechanism of antioxidant protection in *B. longum* in their ecological environment, a growing and a stationary culture of the strain were studied.

## 2. Materials and Methods

### 2.1. Bacterial Strains, Media, Culture Condition 

*B. longum* subsp. *longum* strain GT15 was deposited on 8 June 2012 in the Russian National Collection of Industrial Microorganisms at the Institute of Genetics and Selection of Industrial Microorganisms (VKPM): No. AC-1928. *B. longum* GT15 from the laboratory culture collection was grown at 37 °C in de Mann Rogosa Sharpe (MRS) broth, supplemented with 0.05% L-cysteine HCl, under anaerobic conditions in an anaerobic gas chamber (HiAnaerobic System Mark III, HiMedia, India). 

### 2.2. Treatment of B. longum Culture by Oxidative Stress

To study oxidative stress responses, the culture of bifidobacteria was treated with 1 mM of H_2_O_2_ for 2 h and incubated on a shaker for 2 and 4 h at 37 °C. 

A culture grown in a MRS broth with cysteine HCl to the middle of the exponential growth phase (OD600—0.5–0.6) under anaerobic conditions at 37 °C was divided into two portions: control and experiment. For the control, a portion of the culture was not subjected to oxidative stress, and samples were taken after 2 h and 4 h of incubation under anaerobic conditions. For the experiment, after centrifugation at 28 °C and 7500× *g*, removal of the medium with cysteine HCl, and the addition of an equivalent volume of MRS broth, the culture was divided into three parts: two parts were put on a shaker and incubated at 250 rpm and 37 °C, and in the other part, H_2_O_2_ was added up to 1 mM and was incubated at 37 °C in aerobic conditions. Samples were taken after 2 h and 4 h of incubation on the shaker and after 2 h from the culture with 1 mM of H_2_O_2_ added.

### 2.3. Total RNA Isolation and Sequencing

Cells grown in MRS broth (10^8^ cfu/mL) were harvested (centrifugation for 1 min at 12,000× *g*) and the pellet was resuspended in 100 µL of TE buffer (30 mM/L Trise HCl, 1 mM/L ethylenediamine tetraacetic acid (EDTA), pH 8.0), supplemented with lysozyme (20 mg/mL) and incubated for 10 min at 37 °C. Then, 350 µL of lysis/binding buffer (4.5 mol/1 guanidine/HC1, 50 mM/L TrisHCI, 30% Triton X-100, pH 6.6) was added to the solution. The cells were disrupted by vortexing with silica beads (150–212 mm diameter). The RNA extraction process was carried out according to the RNeasy Mini Kit (Qiagen, Germantown, MD, USA) manufacturer’s guidelines. DNase I degraded the double-stranded and single-stranded DNA present in the RNA samples. The purity of the RNA samples was tested by NanoDrop^TM^. Using the Agilent 2100 Bioanalyzer (Agilent RNA 6000 Nano Kit), the total RNA samples’ concentration, RIN, 23S/16S, and size were detected. Total RNA samples were treated with Vazyme Ribo-off rRNA Depletion Kit Bacteria to deplete rRNA. The RNA molecules were fragmented into small pieces using a fragmentation reagent. The synthesized cDNA was subjected to end-repair and was then 3′ adenylated. Adapters were ligated to the ends of these 3′ adenylated cDNA fragments. PCR was performed to amplify the cDNA fragments with adapters from the previous step. The PCR products were purified with the XP beads and dissolved in TE solution. The libraries were assessed for quality and quantity by two methods: checking for the distribution of the fragments’ size using the Agilent 2100 bioanalyzer and quantifying the library using real-time quantitative PCR (qPCR) (TaqMan Probe). The double-stranded PCR products were heat-denatured and circularized by the splint oligo sequence. The single-strand circle DNA (ssCir DNA) were formatted as the final library. The qualified libraries were displayed on cBot to generate the cluster on the flow cell, and the amplified flow cells were paired-end sequenced on the DNBSEQ System (BG I). The sequenced transcriptome was analyzed for quality errors using FastQCv0.11.5 [15]. The sequencing quality was improved using Trimmomaticv0.39 [16].

### 2.4. Whole Transcriptome Analysis

The whole transcriptome analysis of the sequenced samples was conducted using the following workflow: First, the reference genomes and their annotations were downloaded from the Ref Seq database by the corresponding ID GCF_000772485.1 for *B. longum* subsp. *longum* GT15. Next, hisat2 v2.2.1 [17] was used for mapping of the quality-checked reads of the sequenced samples. The mappings were filtered using the SAM tools package v1.10 [18]. The read counts were assessed using HTSeq-count v2.0.2 [19].

### 2.5. Forming a Reference Catalog of Orthologs of Genes of Antioxidant Function 

For the analysis of the genes responsible for the antioxidative properties, reference catalogs of amino acid sequences were assembled for the *Bifidobacterium* genus. For that purpose, the following algorithm was used: First, the list of antioxidants produced by the bacteria of the genus was constructed using published materials, including research papers, curated databases, etc. Next, the reference amino acid sequences for the genes encoding enzymes responsible for the production and metabolism of the chosen antioxidants were searched for. Only sequences experimentally approved in the research papers were collected. After that, these reference sequences were used for the search of the orthologs in other species of the corresponding genus using BLAST [20] and the NCBI (NCBI Resource Coordinators, Database Resources, Protein Database 2022). Thus, the catalog of the orthologs was constructed.

### 2.6. Genetic Analysis of Genomes of B. longum Genera

For the analysis of the distribution of the genes from the catalog, concerning *B. longum,* the published genomes of this species were obtained from the NCBI Assembly database. Only genomes with the assembly status ‘complete’ were used. Overall, genomes of 54 strains were analyzed. For identification of the homologs of the genes from the reference catalog, the BlastX program was used. The thresholds for the alignments’ filtering were a minimal identity of 60%, and a minimal relative alignment length of 80%.

### 2.7. Analysis of Expression Levels of the Genes Responsible for Antioxidative Properties

The expression levels of the genes responsible for the antioxidative properties in the sequenced strain were analyzed using the reference catalog. The transcriptome reads were mapped on the amino acid sequences using DIAMOND v2.0.13 [21]. The hits were filtered using custom scripts written in Perl according to the following thresholds: identity ≥ 60% and relative alignment length ≥ 90%. Multiple alignments were also filtered. After that, the number of reads aligned to each gene was counted and normalized.

### 2.8. Sample Preparation for Proteomic Analysis

After exposure to H_2_O_2_ or O_2_, the bacterial cells were collected by centrifugation at 5700× *g* for 30 min at 40 °C. The cell pellets were washed three times in PBS solution (pH 7.4), containing 1 mM of PMSF. The PBS solution was heated at 95 °C for 20 min, after which the cell pellets were resuspended in the PBS solution at a ratio of 1:10 and incubated at 95 °C for 10 min. Cells were broken using ultrasound disintegration with a VibraCell™ Ultrasonic Processor (Sonics, USA). The processing mode consisted of an 80% amplitude, 15 s of sonication, and 10 s intervals between sonications for a total period of 30–45 min at 4 °C. Cellular debris was centrifuged at 25,000× *g* for 20 min at 40 °C. Protein preparations were kept at −20 °C. The concentration of the isolated proteins was measured by a Qubit fluorimeter (Invitrogen, Carlsbad, CA, USA). 

### 2.9. Digestion of Proteins with Trypsin in Solution 

Digestion of the proteins and desalting of the peptides were performed as described previously [22]. Briefly, cells were lysed by heating for 10 min at 95 °C in a 100 mM TRIS buffer, pH 8.5, containing 1% sodium deoxycholate (SDC), 10 mM of TCEP, and 20 mM of 2-chloroacetamide. After cooling the sample to room temperature, the equal volume of trypsin solution in 100 mM of TRIS, pH 8.5, was added in a 1:100 (*w*/*w*) ratio. After overnight digestion, the sample was acidified by 1% TFA, and a volume containing 20 μg of peptides was desalted using SDB-RPS StageTips. The eluted material was vacuum-dried and stored at −80 °C.

### 2.10. Liquid Chromatography and Mass Spectrometry 

The LC-MS/MS analyses of the peptides were performed with an Ultimate 3000 Nano LC System (Thermo Fisher Scientific Inc., Waltham, MA, USA) coupled to the Q Exactive Plus Orbitrap mass spectrometer (Thermo Fisher Scientific Inc., Waltham, MA, USA). The peptide solution in 2% acetonitrile/0.1% TFA was loaded into a home-made trap-column, of 50 × 0.1 mm, packed with Inertsil ODS3 3 µm (GL Sciences Inc., Fukushima, Japan) at 4 µL/min and separated at room temperature in a home-made fused silica column of 300 × 0.1 mm, packed with Reprosil PUR C18AQ 1.9 (Dr. Maisch HPLC GmbH, Ammerbuch, Germany) into an emitter, as described earlier [23]. Solution A (0.1% formic acid in LC-MS-grade water) and solution B (80% acetonitrile (*v*/*v*), 19.9% LC-MS-grade water, 0.1% formic acid) were used for gradient LC separation. Peptides were eluted from the column with a linear gradient of: 3–40% B for 58 min, 40–60% B for 4 min, 60% B for 3 min, 60–99% B for 0.1 min, 99% B for 10 min, and 99–2% B for 0.1 min at a flow rate of 500 nl/min. MS1 scan parameters were as follows: resolution—70,000, scan range—350–1600, max injection time—35 ms, AGC target—3 × 10^6^, isolation window—1.4 *m*/*z*, preferred peptide match and isotope exclusion, and dynamic exclusion—30 s. MS2 fragmentation parameters were as follows: HCD mode with collision energy of 30%, resolution—17,500, max injection time—80 ms, AGC target—1 × 10^5^, charge exclusion—unassigned, 1, >7. 

### 2.11. Proteomic Data Analysis

The MS raw files were analyzed by Peaks Studio 10.0 (Bioinformatics Solutions Inc., Waterloo, ON, Canada) [24]. Identification of proteins was performed by searching against the *Bifidobacterium longum* subsp. *longum* GT15 Uniprot FASTA database version of 19 September 2022, with carbamidomethyl Cys as a fixed modification and deamidation Asn/Gln and Met oxidation as variable modifications. The false discovery rate for peptide–spectrum matches was determined by searching a reverse database and was set to 0.01. Enzyme specificity was set as C-terminal to arginine and lysine, and a maximum of two missed cleavages were allowed in the database search. Peptide identification was performed with an allowed initial precursor mass deviation of up to 10 ppm and an allowed fragment mass deviation of 0.05 Da. 

A quantitative comparison of the relative content of proteins in the different groups of samples was carried out by a label-free quantification method (top 3 peptides) using the Peaks Studio 10.0 program. The statistical significance of the observed differences was assessed using the PEAKS Q method built into the program.

### 2.12. DPPH^•^ Radical Scavenging Assay

The ability of the samples to donate an electron and to scavenge 2,2-diphenyl-1-picrylhydrazyl (DPPH) radicals by electron donation was determined by the slightly modified (by Mihaylova et al. [25]) method of Brand-Williams et al. [26]. A freshly prepared 4 × 10^−4^ M solution of DPPH was mixed with the samples in a ratio of 2:0.5 (*v*/*v*). After incubation at room temperature for 30 min, the absorption was measured at 517 nm. The DPPH radical scavenging activity is presented as a function of the concentration of Trolox and is defined as the concentration of Trolox with equivalent antioxidant activity, expressed as μM TE/g for the cell biomass and μM TE/mL for the culture liquid.

### 2.13. ABTS^•+^ Radical Scavenging Assay

The radical scavenging activity of the samples toward 2,2′-azino-bis(3-ethylbenzothiazoline-6-sulfonic acid) (ABTS^•+^) was evaluated as described by Re et al. [27]. The ABTS radical cation (ABTS^•+^) was produced by reacting ABTS stock solution (7 mM) with 2.45 mM of potassium persulfate (final concentration) and allowing the mixture to stand in the dark at room temperature for 12–16 h before use. Afterward, the ABTS^•+^ solution was diluted with ethanol to an absorbance of 0.7 ± 0.02 at 734 nm and equilibrated at 30 °C. The reaction mixture consists of 1.0 mL of diluted ABTS^•+^ solution added to 0.01 mL of samples and was incubated at 30 °C for 6 min. Afterward, the absorbance reading was taken at 734 nm and 30 °C. The results are expressed as μM TE/g for the cell biomass and μM TE/mL for the culture liquid.

### 2.14. Ferric-Reducing Antioxidant Power (FRAP) Assay

The FRAP assay was carried out according to the procedure of Benzie and Strain [28], with slight modifications. The FRAP reagent was prepared fresh daily and was warmed to 37 °C prior to use. Then, 150 µL of sample was allowed to react with 2850 µL of the FRAP reagent for 4 min at 37 °C, and the absorbance was recorded at 593 nm. The results are expressed as μM TE/g for the cell biomass and μM TE/mL for the culture liquid.

### 2.15. Cupric Ion-Reducing Antioxidant Capacity (CUPRAC) Assay

The CUPRAC assay was carried out according to the procedure of Apak et al. [29]. Here, 1 mL of CuCl_2_ solution (1.0 × 10^−2^ M) was mixed with 1 mL of neocuproine methanolic solution (7.5 × 10^−3^ M), 1 mL of CH_3_COONH_4_ buffer solution (pH 7.0), and 0.1 mL of sample, followed by the addition of 1 mL of water (total volume = 4.1 mL), and then mixed well. Absorbance against a reagent blank was measured at 450 nm after 30 min. Trolox was used as a standard, and the results are expressed as μM TE/g for the cell biomass and μM TE/mL for the culture liquid.

### 2.16. Determination of an Intracellular Metabolic Profile

After exposure to H_2_O_2_ and O_2_, the bacterial cells were separated from the nutrient medium through centrifugation of 10 mL of culture medium for 10 min at 6000 min^−1^ and the obtained biomass was washed twice with 0.5% NaCl solution. The biomass was lyophilized using a Biobase BK-FD18P apparatus for 24 h. The biomass was pre-frozen to −40 °C. Lyophilization took place at a working pressure of 25 Pa. The process ended at a post-drying temperature of 30 °C. The resulting lyophilizates were subjected to extraction, where 15 mg of dry biomass was extracted with 1 mL of solvent containing CH_3_OH, CHCl_3_, and water in a ratio of 2.5:1:1. Then, 0.5 mL of distilled water was added to the samples and centrifuged for 10 min at 6000 min^−1^, and 0.5 mL of the supernatant was separated and lyophilized under the same conditions as the biomass. The obtained dry cell extracts were dissolved in pyridine and subjected to derivatization with N-trimethylsilyl-N-methyl trifluoroacetamide (MSTFA) for 30 min at a temperature of 37 °C.

The intracellular metabolites were analyzed with a Trace 1300 GC GC/MS system equipped with an ISQ QD single-quadrupole mass spectrometer (Thermo Fisher Scientific Inc., Waltham, MA, USA), with a TR-5MS column (length, 30 m; id, 0.25 mm; film thickness, 0.25 μm; Thermo Fisher Scientific Inc., Waltham, MA, USA). Helium was used as the mobile phase at a rate of 1 mL/min.

Samples with a volume of 1 µL were injected into the gas chromatograph injector at a temperature of 270 °C and a split flow of 25 mL/min. The analysis started at a temperature of 110 °C, which was held constant for 3 min, then increased to 330 °C at a rate of 10 °C/min and held for 5 min. The ion source and transfer line temperatures were 220 °C and 260 °C, respectively. Mass spectra from each sample were recorded using a scan range of 45 to 650 *m*/*z*.

Cellular metabolites were identified by comparing their mass spectra with those in the National Institute of Standards and Technology database (NIST MS Search 2.2, 2014). SIMCA 14 2015 software was used to create the OPLS-DA and OPLS models (Sartorius AG, Göttingen, Germany).

## 3. Results and Discussion

### 3.1. Genomic and Transcriptomic Data of Antioxidative Response of B. longum GT15

#### 3.1.1. Genes for Antioxidative Response of *B. longum*

First, enzymes and other cells compounds involved in the oxidative stress response of *B. longum* and the genes encoding them were selected and assembled in Appendix A after analysis of the published data. The orthologs to the selected genes were found in the genomes of *B. longum* and a catalog was assembled of their amino acid sequences from the reference literature, as described in Section 2. The final version of the catalog comprised of 203 sequences for 27 enzymes (Appendix A). Then, we analyzed the presence of the genes from the reference catalog in the complete genomes of different strains of the *B. longum* species. The data on the distribution of the genes encoding the following: alkyl hydroperoxide reductase AhpC and AhpF, NADH oxidase, P-type ATPase, DSBA oxidoreductase, dihydroorotate dehydrogenase, glutaredoxin grxC2 (nrdH), class I pyridine nucleotide disulfideoxidoreductase, glutathione import ATP-binding protein GsiA, linoleic acid isomerase, glutathione peroxidase, permease, peroxiredoxin, thioredoxin, thioredoxin domain protein, thioredoxin peroxidase, thioredoxin reductase, and peroxiredoxin, are presented in Appendix A. This table includes data for the genome of the *B. longum* GT15 strain, which indicates the presence of 16 of the studied genes. The distribution of AO genes demonstrates their conservativeness and the general mechanism of AO protection in representatives of the species *B. longum*. However, the strain-specificity of antioxidant activity in bifidobacteria is a known fact [3].

To identify the functionality of the revealed genes and to identify other genes with AO properties, a general transcriptomic analysis of a growing culture of *B. longum* GT15 exposed to oxidative stress was performed.

#### 3.1.2. Transcriptomic Data for Antioxidative Response of *B. longum* GT15 

For transcriptomic analysis, a culture of *B. longum* GT15 in an exponential growth phase was used, which was exposed to hydrogen peroxide for 2 h and oxygen for 2 and 4 h. These conditions were used as approximations to the environment of bifidobacteria in the intestinal tract with possible oxidative stress during inflammatory processes or during the industrial processes of probiotic production. 

Total RNA was isolated from the culture cells selected before (control) and after the exposure to oxidative stress and sequenced on the DNBSEQ System device after the libraries’ preparation. The transcriptome samples of *B. longum* GT15 were sequenced as paired-end reads of 100 bp. The full parameters of the samples are presented in Appendix A. The quality-checking procedures showed that the samples were good, and their sizes did not reduce much after trimming. The overall amount of data was almost evenly distributed across the samples. The transcriptomic reads were aligned at the reference catalog with DIAMOND and filtered as described in Section 2. The numbers of remaining reads used for the estimation of the expression levels for the 5 samples were: 443,467 (control 1), 407,245 (control 2), 460,909 (H_2_O_2_—2 h), 441,397 (O_2_—2 h), and 394,653 (O_2_—4 h) reads. 

The transcriptomic reads were mapped at the *B. longum* GT15 genome using HTSeq-2. The total numbers of reads used for further estimation of the expression levels in the 5 samples were: 27,580,604 (control 1), 29,076,210 (control 2), 24,965,322 (H_2_O_2_—2 h), 28,207,704 (O_2_—2 h), and 27,838,820 (O_2_—4 h) reads.

The total number of genes with identified transcripts was 1974, of which a more than two-fold increase in the levels of transcripts was detected for the genes, as follows: in response to H_2_O_2_ for 190 genes, in response to the action of O_2_ for 2 h for 117 genes, and in response to the action of O_2_ for 4 h for 177 genes (Appendix A).

The genes encoding products with antioxidant action were identified among the transcripts by using the catalog of orthologs (Table 1). Mostly, a more than 2-fold increase of transcript levels was detected for 11 genes after the oxidative stress response (Table 1). The transcripts for the genes encoding glutaredoxin, thioredoxin, thioredoxin_reductase, p-type_ATPase, and dihydroorotate dehydrogenase were upregulated more than 3-fold after H_2_O_2_ and O_2_ exposure. The genes encoding glutaredoxin, thioredoxin, and thioredoxin reductase were also highly upregulated—more than 6-fold, after 60 min of oxygen exposure in the *B. longum* BBMN68 strain [30]. The thioredoxin-dependent reduction system plays an important role in the oxidative stress response by directly reducing H_2_O_2_, scavenging hydroxyl radicals, quenching singlet oxygen, and maintaining the intracellular thiol-disulfide balance [31]. The gene encoding P-type ATPase was highly (more than 5-fold) upregulated in *B. longum* GT15 after the action of all oxidative stresses. It has been reported that zntA1, encoding P-type ATPase, was upregulated 2.01-fold after 60 min of oxygen exposure in strain BBMN68 [30]. P-type ATPase might be involved in taking up Mn^2+^, which then scavenges superoxide anions in bifidobacteria [11]. Mn^2+^ not only replaces superoxide dismutase in scavenging superoxide anions, but it can also scavenge H_2_O_2_ [32]. The gene encoding dihydroorotate dehydrogenase was also highly upregulated in *B. longum* GT15 after H_2_O_2_ (more than 6-fold) and oxygen exposure (more than 8-fold). Dihydroorotate dehydrogenase could be involved in H_2_O_2_ production in highly aerated environments in bifidobacteria [33]. When exposed to oxygen, the transcript for the gene encoding DSBA oxidoreductase was upregulated more than 8-fold (2 h) and 18-fold (4 h), and for the genes encoding thioredoxin domain_protein was upregulated more than 2-fold (2 h) and 8-fold (4 h). The genes encoding NADH_oxidase and thioredoxin_peroxidase were upregulated 3.74- and 6.76-fold, correspondingly, after only 4 h of oxygen exposure in *B. longum* GT15. The NADH oxidase homologue, together with other predicted proteins, decreased oxidative damage in bifidobacteria [9]. The gene encoding Class I pyridine nucleotide-disulfide oxidoreductase was upregulated in the cells of *B. longum* GT15 by more than 9-fold after H_2_O_2_ and more than 2-fold after O_2_ (2 h) exposure, similar to the process in the cells of the *B. longum* BBMN68 strain after oxygen exposure (60 min) [30]. However, an increase in protein levels in the cells of *B. longum* GT15 was observed only after the action of H_2_O_2_. The transcript for the gene encoding linoleic_acid_isomerase was upregulated more than 3-fold after H_2_O_2_ exposure. Conjugated linoleic acid, formed by some types of bifidobacteria, itself does not have AO properties, but its metabolites exhibit the ability to protect cells from harmful oxidative effects [34,35]. The levels of transcripts of the other genes encoding antioxidant products were upregulated by less than 2-fold.

To identify the translation products of the AO genes under the same conditions of oxidative stress, a proteomic analysis of the growing cell culture of *B. longum* GT15 was carried out.

### 3.2. Proteomic Data for Antioxidative Response of B. longum GT15 in Comparison with Transcriptomics Data

For proteomic analysis, a growing culture of *B. longum* GT15 in an exponential growth phase was also used, which was exposed to hydrogen peroxide for 2 h and oxygen for 2 and 4 h. 

Proteins were isolated from the culture cells selected before (control) and after the action of oxidative stress and analyzed using the proteomic approach through liquid chromatography and mass spectrometry. As a result of the proteomic analysis, 1270 proteins were identified. The volcano slot for proteins of *B. longum* GT15 after inducing a stress response, as seen in Figure 1, shows the difference in levels of proteins after the exposure to various types of oxidative stress. Under the action of H_2_O_2_, a difference was detected for 37 proteins, under the action of O_2_ for 2 h, a variation was detected for 125 proteins, and under the action of O_2_ for 4 h, for 68 proteins. For the analysis, only increased levels of proteins after exposure to oxidative stress were used. An increase in levels of more than twice was detected for the proteins: in response to H_2_O_2_ for 13 proteins, in response to the action of O_2_ after 2 h for 19 proteins, and in response to the action of O_2_ after 4 h for 14 proteins (Appendix A).

Five known proteins with antioxidant functionality were identified in the stress response (Table 2). The levels of the proteins glutaredoxin and thioredoxin were increased under the action of oxygen, and pyridine nucleotide-disulfide oxidoreductase levels were increased under the action of H_2_O_2_. Proteins ribonucleoside-triphosphate reductase and P-type ATPase more than doubled in the cells of *B. longum* GT15 after the action of oxygen for 2 h. Class_I pyridine nucleotide-disulfide oxidoreductase increased after the addition of H_2_O_2_. 

Thus, the increased levels of both transcripts and the proteins glutaredoxin and thioredoxin in the cells of the *B. longum* GT15 strain after exposure to oxygen indicate that the thioredoxin-dependent AO system may be the major redox homeostasis system in *B. longum* bacteria, which has been indicated in other published studies [30].

Other identified proteins with more than twice increased levels in cells after the action of oxidative stress are involved in the processes of the multiple stress response in the cells, amino acid metabolism, nucleotide and carbohydrate metabolism, and transport. In the cells of *B. longum* GT15, after exposure to oxidative stress, the level of proteins, involved in proper protein folding (Co-chaperonin GroES) and degradation of damaged protein (serine protease); taking part in repair, removing, proofreading, protecting DNA during starvation, and oxidative stress (DNA polymerase sliding clamp subunit, nucleotidyl transferase, DNA topoisomerase); modifying RNA secondary structures or intermolecular RNA interactions, and modulating RNA–protein complexes (DEAD/DEAH box helicase), were increased. In *B. longum* GT15 cells, excessive levels were identified for the proteins: XRE family transcriptional regulator, MerR family transcriptional regulator, LexA repressor, cystathionine beta-synthase, and general stress protein, after oxygen or H_2_O_2_ exposure. These proteins play an important role in the adaptation of bacteria to changing environments and in their response to stress conditions, including oxidative stress. These results are supported by other authors, reporting that chaperones and proteases related to several stress conditions were induced in *B. longum* BBMN68 in response to oxygen [30]. The GroEL/GroES complex is required for proper protein folding and is frequently involved in responses to heat, low pH, and bile-salt stresses in bifidobacteria [36]. 

Several genes encoding proteases and peptidases were upregulated in *B. longum* BBMN68 after 60 min of exposure to oxygen [30]. These proteins play a major role in the degradation and turnover of damaged proteins [30]. The SOS response in bacteria is a global regulatory network for DNA-damage repair, produced by reactive forms of oxygen, governed by the repressor LexA and the inducer RecA [37]. In *B. longum* BBMN68, LexA expression was upregulated 3.50-fold after 60 min of oxygen exposure [30]. The transcription elongation factor GreA was increased in the cells after 2 h of exposure to oxygen. GreA aids in adaptation to stressful environments in various bacteria [38].

Interestingly, under the action of hydrogen peroxide, the levels of proteins involved in nucleotide (orotate phosphoribosyl transferase, multifunctional fusion protein cytidylate kinase) and carbohydrate metabolism (gluconate kinase and alpha-amylase) were upregulated in *B. longum* GT15. Only the transcript of the gene encoding alpha-amylase was upregulated after exposure to H_2_O_2_. Alpha-amylase is a metal-activated extracellular endo-acting enzyme that randomly hydrolyzes α-1,4 glycosidic linkages of starch. 

Several proteins, involved in the metabolism and transport of amino acids—cystathionine gamma-synthase, chorismate synthase, and amino acid ABC transporter—were increased in the cells of *B. longum* GT15 after H_2_O_2_ and oxygen exposure compared to control cells. Cystathionine gamma-synthase catalyzes the committed step of de novo methionine biosynthesis. Methionine in proteins acts as an endogenous antioxidant and defends cells against oxidative stress [39]. Chorismate synthase participates in phenylalanine, tyrosine, and tryptophan biosynthesis. Tyrosine and tryptophan residues, accumulated in the transmembrane domains, especially in the region of the highest lipid density, perform vital antioxidant functions inside lipid bilayers and protect cells from oxidative destruction [40]. The ABC transporter is a member of a large family of ATP-binding proteins that transport a variety of molecules across biological membranes. It is possible that bifidobacteria cells require multiple amino acid transporters to fulfill their metabolic requirements during oxidative stress. 

Three uncharacterized proteins were upregulated in *B. longum* GT15 after oxygen exposure at the level of transcription and translation. In the future, we need to pay attention to the characterization of the functions of these proteins.

Thus, the obtained data indicate that *B. longum* GT15, in addition to the known mechanisms of protection against OS, also uses the mechanism of common stress response to ensure its survival. 

Not all proteins with an increased level of translation showed an increase in the level of transcripts, and vice versa. An increase in levels of more than twice was detected for the proteins and their transcripts: in response to H_2_O_2_ for 8 proteins, in response to the action of O_2_ after 2 h for 14 proteins, and in response to the action of O_2_ after 4 h for 13 proteins (Appendix A). An insignificant difference in the expression levels for a number of genes and proteins can be explained by their instability when using the described methods for proteomic or transcriptomic analysis. Part of the proteins may be secreted from the cell and not considered when determining the total cellular proteins. The regulation of a number of proteins can be carried out post-transcriptionally. Oxidants can suppress translation. Our results demonstrate that the reaction of gene transcription in response to OS was much stronger than protein synthesis, as it has been shown in other studies [41].

### 3.3. Metabolomic Data for Antioxidative Response of B. longum GT15

#### 3.3.1. Antioxidant Activity of *B. longum* GT15

The in vitro antioxidant activity of the culture fluid and cell biomass of *B. longum* GT15 was investigated when hydrogen peroxide and oxygen were applied as stress agents (Table 3). Since the metabolites synthesized by the strain can be both exo and endo, both the cell biomass and the culture fluid were examined to evaluate the antioxidant potential. When examining the culture fluid, the reaction to oxygen, assessed at 4 h intervals, showed less sensitivity compared to hydrogen peroxide, which provoked a decrease in the antioxidant potential. The antioxidant activity restored at the end of oxidative stress is reflected in a better potential when oxygen is applied as a stress factor.

From the obtained results, it is clear that the effect of oxygen compared to hydrogen peroxide is weaker and the decrease in the activity of the culture liquid is greater compared to that under the influence of hydrogen peroxide.

As for the biomass of the *B. longum* GT15 strain, the observed antioxidant effect was different. In general, the effect of hydrogen peroxide provokes an increase in the antioxidant activity of the cellular biomass, which can be attributed to internal metabolites and a systemic stress response.

The results of the four in vitro methods indicate the need to perform more than three assays and to extend the studies with an in vivo assay if possible/in the future.

Amaretti et al. [42] investigated the antioxidant properties of various *Lactobacillus*, *Bifidobacterium*, *Lactococcus,* and *Streptococcus thermophiles* and reported that the antioxidant mechanism and degree of antioxidant activity were specific for each individual bacterial strain.

As is known, oxidative stress is a condition that occurs as a result of an imbalance of the internal antioxidant–prooxidant system of the cell, which can lead to apoptosis and cell death. In this regard, the consumption of probiotic strains with antioxidant potential is considered beneficial to human health due to the reduction of oxidative damage [43].

The use of probiotics promotes hemostasis, improves immune responses, and prevents many diseases caused by oxidation in the host, so the interest in investigating less studied potential strains is worthwhile [43].

The outcomes found in the present study (Table 3) can be attributed to the established antioxidant enzymes and the above-mentioned thioredoxin-dependent antioxidant system.

The differences in the responses to oxidative stress could be explained by the different responses in the transcriptomic analysis [44]. As mentioned above, a high level of transcripts of oxidative factors was observed for genes encoding glutaredoxin, thioredoxin, and thioredoxin reductase. The effect of both oxidants leads to a different response in terms of the transcriptome and the identified metabolites.

The differential response to oxidizing agents and the results of the validated proteomic and transcriptomic analyses were consistent with the in vitro antioxidant potential of the cell biomass and culture fluid of *B. longum* GT15.

#### 3.3.2. Metabolic Fingerprinting and Data Analysis

Metabolomic analyses and their correlation with the antioxidant activity of the cells would also provide adequate information about the cellular response as a result of oxidative stress. The cell biomass of *B. longum* GT15 after the application of oxidative stress by H_2_O_2_ and O_2_ was subjected to metabolic fingerprinting. The resulting fingerprints of the intracellular low-molecular-weight metabolites were analyzed with the software SIMCA 14. As input data for OPLS-DA (orthogonal projections to latent structures discriminant analysis) and OPLS (orthogonal projections to latent structures), we used the complete metabolic fingerprints for each sample.

The OPLS-DA method allowed distinct separation between the control (cells without oxidative stress) and the samples subjected to stress by oxygen and hydrogen peroxide (Figure 2). The first component presented 31.2% of the total variance and showed the separation between the three groups. The second component presented 11.8% of the total variance and separated the control from the samples subjected to oxidative stress.

OPLS regression models show how the obtained metabolomic fingerprints correlate to the data from conventional methods for antioxidant activity analysis. If the correlation is high enough, GC-MS fingerprints (a fast and reliable method) can be used to determine the antioxidant activity. The results from the OPLS models are presented in Figure 3 and Appendix A, and they show varying correlations between the predicted and the observed results. The best correlation was observed for the CUPRAC method for the determination of antioxidant activity (R^2^ = 0.8777), with a relatively low estimation error (RMSEE = 0.0530688) and cross-validation error (RMSEcv = 0.0920637) (Figure 3). The results from the modeling of the ABTS method data showed a smaller correlation (R^2^ = 0.7248) and a higher estimation error (RMSEE = 0.096024) and cross-validation error (RMSEcv = 0.127565) (Appendix A). For the DPPH and FRAP methods, the OPLS models had a much lower correlation with the experimental data (Appendix A)—R^2^ = 0.4638 and 0.3335, respectively.

The results obtained from the four applied methods for the evaluation of antioxidant activity are difficult to compare due to the different mechanisms, redox potentials, pH values, solvent dependencies, etc., of the various analyses [45]. However, the CUPRAC assay stands out as a widely used method to evaluate the antioxidant potential of biological fluids because it is performed at a realistic pH value close to physiological pH and a favorable redox potential, but is also applicable to lipophilic antioxidants, as well as hydrophilic ones [46]. In our investigation, the results from the CUPRAC method correlated best to the OPLS model and they showed the lowest estimation and cross-validation errors.

The OPLS-DA showed good discrimination between different phenotypes, and a larger fingerprint database would potentially allow identifying strains with high antioxidant activity. The differential response to oxidizing agents and the results of the validated metabolomic analyses were consistent with the in vitro antioxidant potential of the cell biomass and culture fluid of *B. longum* GT15.

## Figures and Tables

**Figure 1 foods-12-03356-f001:**
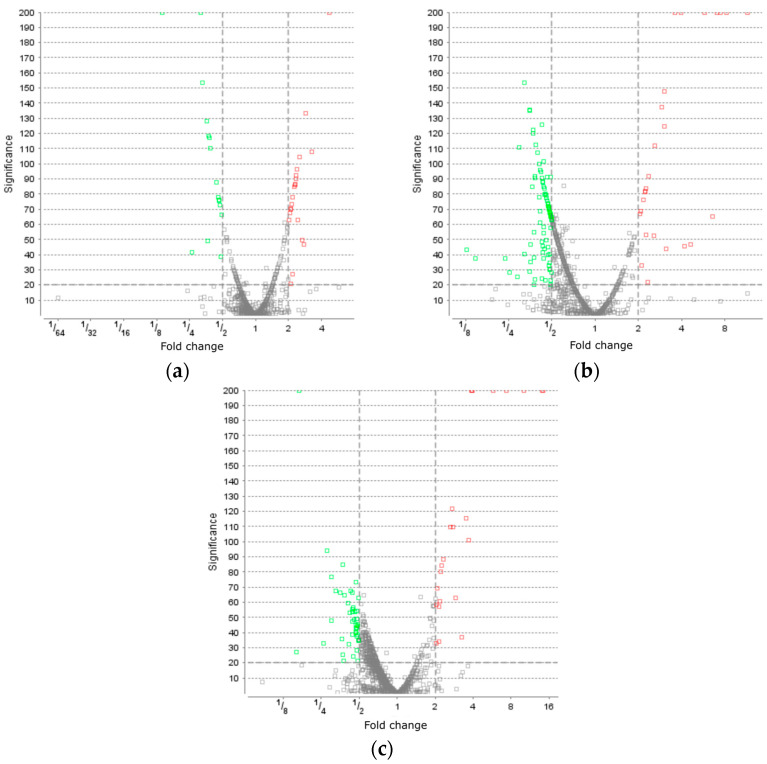
The volcano plot for proteins of *B. longum* GT15 after: (**a**) H_2_O_2_ induced a stress response, (**b**) O_2_ induced a stress response during 2 h of incubation, and (**c**) O_2_ induced a stress response during 4 h of incubation. LFQ quantification was performed via the ‘top 3 peptide’ method, and significance was estimated by the Peaks Q method, built into Peaks Studio 10.0.

**Figure 2 foods-12-03356-f002:**
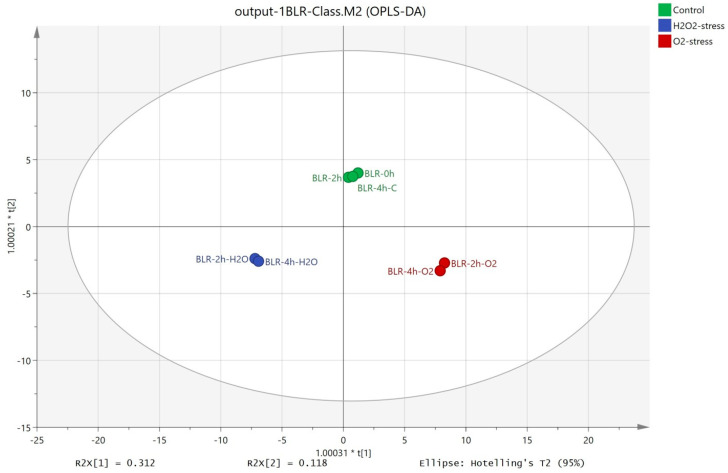
OPLS-DA of the intracellular metabolites extracted from *B. longum* GT15 cells subjected to oxidative stress.

**Figure 3 foods-12-03356-f003:**
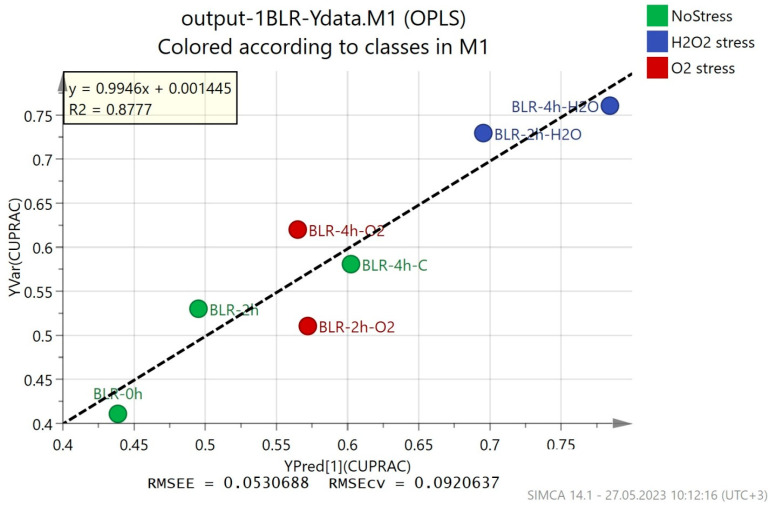
OPLS regression models on metabolic fingerprints of *B. longum* GT15 cells and their antioxidant activity (CUPRAC method).

**Table 1 foods-12-03356-t001:** Changes in transcript levels of genes encoding enzymes of the antioxidative response of the *Bifidobacterium longum* subsp. *longum* GT15 before and after the action of oxidative stress.

No.	Product of the Gene	Locus_tag	* H_2_O_2_/ Control	O_2_ 2 h/ Control	O_2_ 4 h/ Control
1	Glutaredoxin	BLGT_RS07125	3.5	11.4	28.6
2	Thioredoxin	BLGT_RS02480	13.07	6.8	7.75
3	DSBA_oxidoreductase	BLGT_RS00210	1.24	8.55	18.35
4	Thioredoxin_reductase	BLGT_RS00265	5.47	4.35	4.35
5	P-type_ATPase	BLGT_RS01395	18.11	5.88	9.80
6	Thioredoxin_domain_protein	BLGT_RS01710	1.42	2.48	8.05
7	Thioredoxin_peroxidase	BLGT_RS04835	1.54	1.37	6.76
8	Linoleic_acid_isomerase	BLGT_RS06870	3.61	0.99	1.88
9	Class_I_pyridine_nucleotide-disulfideoxidoreductase	BLGT_RS08540	9.02	2.79	1.34
10	NADH_oxidase	BLGT_RS09280	0.74	0.72	3.74
11	Dihydroorotate dehydrogenase	BLGT_RS02540	6.59	9.02	8.68

*: Fold ratio to the abundance across different samples before and after oxidative stress.

**Table 2 foods-12-03356-t002:** Comparative data from proteomic and transcriptomic analyses of cells of *B. longum* GT15 in the exponential growth phase before and after exposure to H_2_O_2_ and O_2_.

No.	Product of the Gene	Locus_Tag	H_2_O_2_ 2 h/ Control	O_2_ 2 h/Control	O_2_ 4 h/ Control	COG *** Category
*Oxidative response*						
1	Glutaredoxin	BLGT_RS07125	NS 3.5	2.89 * 11.4 **	2.15 28.6	O
2	Thioredoxin	BLGT_RS02480	NS 13.07	2.59 6.8	2.71 7.75	O
3	Class_I pyridine nucleotide-disulfide oxidoreductase	BLGT_RS08540	2.17 9.01	NS 2.79	NS 1.32	C
4	Ribonucleoside-triphosphate reductase	BLGT_RS07845	NS 0.84	2.17 0.68	NS 1.32	H
5	P-type ATPase	BLGT_RS01395	NS 17.40	2.24 5.65	NS 9.48	RP
*Stress response*						
6	Cystathionine gamma-synthase	BLGT_RS03020	2.63 11.43	5.78 2.86	5.74 2.96	E
7	Co-chaperonin GroES	groES	NS 35.69	NS 4.84	2.18 46.57	O
8	Transcription elongation factor GreA	BLGT_RS03800	NS 0.73	2.23 4.47	NS 5.28	K
9	DEAD/DEAH box helicase	BLGT_RS07930	2.63 2.7	2.27 1.26	NS 1.04	L
10	LexA repressor	BLGT_RS06740	NS 1.19	2.33 1.38	2.93 3.21	KT
11	General stress protein	BLGT_RS07865	2.77 1.89	NS 1.27	NS 2.3	R
12	DNA topoisomerase	BLGT_RS06115	2.17 0.95	NS 1.1	NS 1.19	L
13	Nucleotidyl transferase	BLGT_RS07810	NS 3.5	4.65 3.9	3.25 5	-
14	DNA polymerase sliding clamp subunit	BLGT_RS00420	NS 7.84	7.47 11.32	14.46 43.6	VP
*Nucleotide metabolism*						
15	Orotate phosphoribosyl transferase	BLGT_RS04975	2.27 0.77	NS 1.02	NS 1.23	-
16	Multifunctional fusion protein cytidylate kinase	BLGT_RS05305	3.84 1.0	NS 0.63	NS 0.29	J
17	Ribonucleoside-diphosphate reductase subunit beta	BLGT_RS00155	NS 0.7	2.08 1.98	NS 2.1	F
*Amino acid metabolism and transport*						
18	Cystathionine beta-synthase	BLGT_RS03015	3.22 6.62	7.00 1.69	7.38 1.99	E
19	Chorismate synthase	BLGT_RS04580	3.12 0.45	3.10 0.85	NS 1.17	E
20	Amino acid ABC transporter	BLGT_RS07135	NS 3.6	3.96 11	3.89 3.6	E
*Carbohydrate metabolism*						
21	Gluconate kinase	BLGT_RS06825	2.77 0.67	NS 0.70	NS 1.04	G
22	Alpha-amylase	BLGT_RS00840	2.13 13.0	NS 1.21	NS 1.35	G
*Regulation of transcription and biosynthesis*						
23	XRE family transcriptional regulator	BLGT_RS06195	7.14 17.1	11.55 6.1	14.17 25.9	K
24	MerR family transcriptional regulator	BLGT_RS00895	2.08 9.34	NS 1.72	NS 4.81	K
*Transport*						
25	Membrane protein	BLGT_RS07320	NS 0.62	8.21 1.9	10.07 1.27	H
26	Cation transporter	BLGT_RS06745	NS 1.61	NS 2.42	2.06 3.0	P
*Uncharacterized proteins*						
27	Uncharacterized protein	BLGT_RS00505	NS 1.05	3.02 2.42	2.31 4.12	-
28	Uncharacterized protein	BLGT_RS06485	NS 6.4	3.04 5.3	3.50 13	-
29	Uncharacterized protein	BLGT_RS07070	NS 5.4	3.58 7.7	3.94 17.8	-

*—Fold ratio to the average abundance of proteins across different samples before and after oxidative stress; **—fold ratio to the abundance of transcripts across different samples before and after oxidative stress; *** COGs—Database of Clusters of Orthologous Genes.

**Table 3 foods-12-03356-t003:** In vitro antioxidant activity of culture liquid (µMTE/mL) and cell biomass (µMTE/g) of *B. longum* GT15 under oxidative stress.

Sample/Assay *	DPPH	ABTS	FRAP	CUPRAC
*Culture liquid*				
Control **—0 h	0.61 ± 0.01	6.50 ± 0.04	1.26 ± 0.03	2.04 ± 0.02
Control—2 h	0.82 ± 0.00	7.30.03	1.94 ± 0.02	2.43 ± 0.04
Control—4 h	0.75 ± 0.01	6.15 ± 0.07	0.92 ± 0.02	2.10 ± 0.02
O_2_—0 h	0.61 ± 0.01	6.50 ± 0.04	1.26 ± 0.03	2.04 ± 0.02
O_2_—2 h	0.81 ± 0.0	6.69 ± 0.02	1.44 ± 0.02	2.25 ± 0.01
O_2_—4 h	0.63 ± 0.01	6.38 ± 0.08	1.33 ± 0.01	2.40 ± 0.06
H_2_O_2_—0 h	0.53 ± 0.00	6.39 ± 0.04	0.74 ± 0.02	2.93 ± 0.05
H_2_O_2_—2 h	0.50 ± 0.01	6.49 ± 0.08	0.92 ± 0.02	2.89 ± 0.04
*Cell biomass*				
Control *—0 h	0.06 ± 0.00	0.87 ± 0.02	0.084 ± 0.003	0.41 ± 0.01
Control—2 h	0.08 ± 0.00	0.91 ± 0.01	0.048 ± 0.000	0.53 ± 0.01
Control—4 h	0.07 ± 0.00	1.06 ± 0.016	0.09 ± 0.00	0.58 ± 0.02
O_2_—0 h	0.06 ± 0.00	0.87 ± 0.02	0.084 ± 0.003	0.41 ± 0.01
O_2_—2 h	0.10 ± 0.00	1.15 ± 0.03	0.08 ± 0.00	0.51 ± 0.01
O_2_—4 h	0.08 ± 0.00	0.94 ± 0.02	0.21 ± 0.01	0.62 ± 0.01
H_2_O_2_—0 h	0.06 ± 0.00	0.87 ± 0.02	0.084 ± 0.003	0.41 ± 0.01
H_2_O_2_—2 h	0.06 ± 0.00	1.03 ± 0.02	0.09 ± 0.00	0.73 ± 0.02

*—In vitro antioxidant activity assessed against 2,2-diphenyl-1-picrylhydrazyl radical activity (DPPH), 2,2′-azino-bis(3-ethylbenzothiazoline-6-sulfonic acid) radical activity (ABTS), ferric-reducing antioxidant power assays (FRAP), and copper ion reducing antioxidant capacity (CUPRAC) assays. **—*B. longum* GT15 without oxidative stress exposure.

## Data Availability

The datasets used and/or analyzed during the current study are available from the corresponding author upon reasonable request.

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
