# Peer review of "Oxidative Stress Response of Probiotic Strain Bifidobacterium longum subsp. longum GT15"

_foods, 2023, doi:10.3390/foods12183356_

Round 1
Reviewer 1 Report
The manuscript "Oxidative stress response of probiotic strain Bifidobacterium longum subsp. longum GT15" is well written and interesting.
However, some aspects of the methodology applied should be clarified and also some aspects of the Results and Discussion sections can be improved.
1) GT15 strain has been defined as probiotic, can you add or highlight in the text the literature that sustains it and the results showing its probiotic properties?
2) Did you perform replicates of transcriptomic and proteomic analysis for each condition?
For example, how many replicates of RNA were extracted from independent cultures and sequenced? This comment is also in light of the results depicted in lines 523-525. Modify the text accordingly.
3) The whole genome of Bifidobacterium longum subsp. longum GT15 was sequenced, thus in paragraph 2.4, "First, the reference genomes and their annotations were downloaded from the Ref Seq database by the corresponding IDs: GCF_000772485.1 for B. longum subsp. longum GT15.", why did you refer to genomes and not a single genome?
4) Lines 486-488: revise the sentences or punctuation.
5) Lines 516-518: are there any indications from the comparison of the uncharacterized proteins with respect to database protein at least from a predictive point of view?
6) Is the Discussion section missing or is it included in the Result section? Modify the text accordingly adding.
7) The headline of Paragraph 2.4 is not separated from the previous one by a space.
Author Response
We would like to thank Reviewer 1 for the remarks on the manuscript and we believe that the corrections we`ve made will improve the quality of the presented paper. Our response to the reviewer`s comments is presented below:
The text was modified according to the remarks.
Response to point 3. Yes, we've used the single assembly of the genome. It is a typo
Response to point 5. Alignments of the first sequence (BLGT_RS06485) with BLASTx against Bacteria (taxid:2) show only "hypothetical protein" results. For the second sequence (BLGT_RS07070) the description in the annotation states: DUF3107 domain-containing protein
Reviewer 2 Report
This manuscript focuses on the effect of Lactobacillus combined with tryptophan on intestinal barrier function. The article shows that the antioxidant protection mechanism of Bifidobacterium longum subsp. longum GT15 was explored by transcriptomic, proteomic, and metabolomic analyses. However, some questions should be corrected or responded to.
A major revision would be required.
The following are the questions in this manuscript:
1. In line 21, bifidobacteria should be italicized and its initial letter should be capitalized. Please check and correct the entire manuscript.
2. In line 25, "Data genomic analysis" should be "Genomic data analysis", and similarly, "Data transcriptomic analysis" in line 27 and "Data proteomic analysis" in line 29.
3. The abstract section lacks a conclusive description.
4. In line 48, "In the gut microbiota of vaginally delivered breast-fed infants the amounts of bifidobacterial reaches 90%." should be "In the gut microbiota of vaginally delivered breast-fed infants, the amounts of bifidobacterial reaches 90%."
5. In line 48, "Actinobacteria" should not be italicized.
6. In line 74, the first occurrence of an abbreviation should be preceded by the full name, e.g., "OS (oxidative stress)" should read "oxidative stress (OS)"; please correct this throughout the manuscript.
7. Key genes screened by transcriptomics should be validated using qPCR.
8. Why was the metabolome done using GC-MS and not LC-MS? Also, differential metabolites need to be validated using a targeted metabolome.
9. Please check carefully for spelling errors and grammatical mistakes in the whole text. It would be best to have a native English speaker do the language modification.
Author Response
We would like to thank Reviewer 2 for the remarks on the manuscript. We hope that the corrections will benefit the quality of the manuscript. The response to the Reviewer`s comments is presented below:
The remarks were taken into account.
Respond to point 8: In the development of metabolomics, GC-MS was preferred because of the narrow peaks it produced. This method also produces metabolomic fingerprints at standard ionization energy of 70 eV and this allows the use of libraries for metabolite identification. Lately, LC-MS has improved and is starting to be used in metabolomic analysis, but since we started working in this field early, we have more experience with GC-MS. Regarding the validation of the differential metabolites using a targeted metabolome – we are not sure what the reviewer means. The method for metabolomic fingerprint analysis does not need validation. We are using the largest available library - NIST Mass Spectral Library - to identify metabolites, but metabolic fingerprinting does not require the identification of individual metabolites. It uses the entire metabolomic fingerprints for data analysis.
Reviewer 3 Report
Averina et. al. used transcriptomics, proteomics and metabolomics approaches to investigate the response of Bifidobacterium longum GT15 to oxidative stresses. They found that this bacterium strain might use thioredoxin and glutaredoxin as the major mechanism to act against oxidative stress. While this study is interesting and the authors found a lot of information from the analyses of their data, the results were not well presented and interpreted. The English language need significantly improve to clarify the manuscript, and make it concise and easy to understand. There were also many grammar errors need to be fixed. Specific comments are listed below.
1. Lines 25,27,29: genomic data analysis , transcriptomic data analysis, proteomic data analysis.
2. Line 27: … demonstrated that (among the 16 genes identified?), 11 genes responsible for antioxidative response increased at least two times when the bacterium was exposed to H2O2 and/or oxygen.
3. Lines 32: We also found that proteins presumably involved in global stress response, amino acid metabolism, …… significantly increased responding to oxidative stress.
4. Line 49, during later stage of life, ..
5. Lines 187: Expression levels of genes …
6. Lines: 353, please clarify the sentences.
7. Lines 384, By using …., 11 genes encoding proteins for antioxidant action were identified.
8. A major issue: most of the tables and figures did not indicate any replicates and standard deviations. Please add the information in the caption for each figure or table. Table 3 also needs note to the methods used to collect the data.
9. Table 2: Is there a better way to present the data? For example, use bar charts.
10. Line 528: please rewire this sentence.
11. It is not clear what points Figures 3-6 were trying to claim. I also suggest either combining the figures into one figure with different graphs or showing the most significant one and put other three into supplemental material.
12. Many other English errors are all over the manuscript. Please check throughout.
Overall, the manuscript is difficult to understand. The authors are advised to ask help from native English speakers to rewrite the work. The flow of the manuscript is also very blur. Please think carefully and outline the most interesting findings to tell a story for the Result and Discussion section. The manuscript would be improved by presenting predicted metabolic pathways (probably from KEGG or other databases) showing the relevant proteins counteracting oxidative stress.
As mentioned above, the manuscript has many grammar errors and is difficult to understand. The authors are advised to ask help from native English speakers to rewrite the work. The flow of the manuscript is also very blur.
Author Response
We would like to thank Reviewer 3 for the remarks on the manuscript. Our response to reviewer`s comment is the following:
The remarks were taken into account and the text was changed according to all of them.
Respond to point 9: It will be difficult to show this table as a bar chart. It will include almost 300 bars, which is obviously far from informative. Theoretically, I could divide this into 8 figures (one per each protein group), so we can present them as parts of one big figure, but still won't be better in my opinion. For example, the "Stress response" group would include 54 bars. I say we keep the data table.
Respond to point 11. OPLS models (depicted in figures 3-6) show how the obtained metabolomic fingerprints correlate to the data from conventional methods for antioxidant activity analysis. If the correlation is high enough GC-MS fingerprints (a fast and reliable method) can be used to determine antioxidant activity. In our opinion combining the figures into one figure will reduce the size of the graphs too much to be legible. One figure was selected, and the rest were added to the supplementary materials.
Reviewer 4 Report
Generally, the manuscript includes detailed methodology and results on different approaches; however, in the end the correlations between all these important findings are not sufficiently/convincingly outlined and explained how can be actually useful when designing novel functional products as claimed in introduction.
Line 112: probably incubated” instead of incubation”
Lines 118: delete “)” after7.500 g to make clear the phrase
Author Response
We would like to thank Reviewer 4 for the remarks on the manuscript. We hope that the corrections will benefit the quality of the manuscript. Corrections were made in the manuscript according to the reviewer’s notes.